# Is Personalized Mechanical Thrombectomy Based on Clot Characteristics Feasible? A Radiomics Model Using NCECT to Predict FPE in AIS Patients Undergoing Thromboaspiration

**DOI:** 10.3390/jcm14124027

**Published:** 2025-06-06

**Authors:** Jacobo Porto-Álvarez, Javier Martínez Fernández, Antonio Jesús Mosqueira Martínez, Miguel Blanco Ulla, Susana Arias Rivas, Emilio Rodríguez Castro, Ramón Iglesias Rey, José M. Pumar, Roberto García-Figueiras, Miguel Souto Bayarri

**Affiliations:** 1Department of Radiology, Hospital Clínico Universitario de Santiago de Compostela, 15706 Santiago de Compostela, Spain; jacoporto@hotmail.com (J.P.-Á.); mblancoferrol@gmail.com (M.B.U.); josemanuel.pumar@usc.es (J.M.P.); roberto.garcia.figueiras@sergas.es (R.G.-F.); miguel.souto@usc.es (M.S.B.); 2Department of Neurology, Hospital Clínico Universitario de Santiago de Compostela, 15706 Santiago de Compostela, Spain; susana.arias.rivas@sergas.es (S.A.R.); emiliorcastro@gmail.com (E.R.C.); 3Neuroimaging and Biotechnology Laboratory (NOBEL), Clinical Neurosciences Research Laboratory (LINC), Health Research Institute of Santiago de Compostela (IDIS), 15706 Santiago de Compostela, Spain; ramon.iglesias15@gmail.com

**Keywords:** radiomics, stroke, AIS, thromboaspiration, FPE, TICI

## Abstract

**Background/Objectives**: In patients with acute ischemic stroke (AIS), the first pass effect (FPE) refers to the complete recanalization of an occluded vessel (TICI = 2C/3) with a single thrombectomy attempt. Achieving complete vessel recanalization is associated with better functional outcomes compared to lower reperfusion rates (TICI < 2B). There is no consensus on which thrombectomy technique provides the best recanalization results for AIS patients. Furthermore, there is a paucity of tools available to predict FPE prior to mechanical thrombectomy (MT). The objective of this study is to develop a radiomics model based on brain NCECT to predict which patients are more likely to achieve a FPE with thromboaspiration MT. **Methods**: The thrombi of 91 patients were semi-automatically segmented on NCECT. A total of 1167 radiomic features (RFs) were extracted for each patient. Some clinical data (age, gender, cardiovascular risk factors, smoking or alcohol abuse, clot density and clot laterality) were also collected. **Results**: A LASSO regression analysis identified nine RFs with nonzero coefficients. A logistic regression model for FPE prediction was developed with nine RFs and eight clinical variables. A total of six RFs were found to be statistically associated with FPE. The clinical variables did not demonstrate a statistically significant association with the likelihood of achieving FPE (*p* > 0.05). The prediction of which patients are likely to achieve FPE obtained an AUC, accuracy, sensitivity and specificity of 0.890, 0.813, 0.815 and 0.811, respectively (*p* < 0.05). **Conclusions**: Radiomics can help identify patients who are more likely to achieve FPE with thromboaspiration.

## 1. Introduction

Recanalization of the occluded vessel in acute ischemic stroke (AIS) patients after a mechanical thrombectomy (MT) is measured with the TICI (Thrombolysis In Cerebral Infarction) scale [1]. Classically, successful repermeabilization was defined as a thrombectomy resulting in a TICI ≥ 2B [2]. However, it has been shown that patients who achieve a recanalization TICI 2C or 3 have better clinical outcomes 3 months after thrombectomy than patients who achieve TICI 2B [3]. In patients undergoing mechanical thrombectomy, the first pass effect (FPE) is defined as the complete recanalization (TICI ≥ 2C) of an occluded vessel following a single-pass thrombectomy. It has been demonstrated that FPE is associated with improved clinical outcomes and lower mortality rates in patients with AIS [4,5]. Initially, the FPE was described for stent retriever devices. Subsequent studies evaluated the rate of FPE in patients undergoing stent retriever therapy compared to those undergoing thromboaspiration. The results were found to be similar in both groups in randomized trials [6,7]. Furthermore, an association has been observed between the molecular composition of the thrombus and the probability of achieving an FPE with MT [8,9]. Moreover, research has been conducted that demonstrates a correlation between the molecular composition of thrombi and their radiological appearance, as well as the correlation between the radiological appearance and the probability of FPE [10,11,12,13].

In the domain of diagnostic imaging, radiomics has witnessed a surge in interest in recent years. This emerging field involves the use of quantitative analysis techniques to process radiological image data that escapes human visual detection [14]. In its early days, radiomics focused primarily on oncological pathology, but more recently, there have been more publications on radiomic modeling in non-oncological pathology, including in patients with AIS [15,16,17,18]. It has been revealed that radiomic data may vary depending on the molecular composition of the thrombi causing AIS [19]. However, there is a paucity of studies on the prediction of successful repermeabilization in patients undergoing MT using radiomic data from radiological images [20,21]. No studies have been identified in which the objective is to predict FPE with thromboaspiration based on NCECT-based radiomics.

We propose a retrospective study to identify which patients are most likely to achieve an FPE with thromboaspiration using a radiomic model with features obtained from NCECT. Given the relatively rapid nature of thromboaspiration and its cost-effectiveness compared with stent retrievers in cases where an FPE is achieved [22], we believe that such information may be very important in prethrombectomy planning in patients with AIS.

## 2. Materials and Methods

### 2.1. Study Design and Ethical Approval

This retrospective observational study was conducted at the University Hospital of Santiago de Compostela, a public and tertiary-level hospital that serves as a referral center for the treatment of AIS patients, covering a population of 600,000. The study was performed in accordance with the principles of the Declaration of Helsinki of the World Medical Association and approved by the Santiago-Lugo Ethics Committee (code 2023/299).

### 2.2. Patient Selection

A total of 334 mechanical thrombectomies were performed at this hospital between the years 2021 and 2023. All these patients were admitted to the stroke unit of the same hospital and treated according to national society protocols by trained neurologists and neuroradiologists experienced in cerebrovascular disease. Patients presenting to the emergency department with suspected AIS underwent brain NCECT, CT perfusion and CT angiography of the supra-aortic arteries. For recanalization treatment, interventional neuroradiologists employed mechanical thrombectomy by thromboaspiration as a primary technique. The stent retriever and combined (aspiration and stent retriever) techniques were employed as second-intention techniques. The procedures were conducted by neuroradiologists with expertise in both diagnostic and interventional neuroradiology.

Data were collected retrospectively from patients who underwent mechanical thrombectomy at the University Hospital of Santiago de Compostela between 1 January 2020 and 31 May 2023. The inclusion criteria were as follows: (1) patients aged 18 years old or older; (2) brain NCECT with a slice thickness of 0.625 mm; (3) a visible thrombus on NCECT located in the distal internal carotid artery (ICA) or in the middle cerebral artery (MCA) (M1 or proximal M2 segments); (4) treatment with at least the first thrombectomy attempt performed with thromboaspiration. The exclusion criteria were as follows: (1) thrombus in another intracranial artery or patients with tandem occlusion and (2) patients for whom imaging studies were conducted in an external hospital. Intravenous tissue plasminogen activator (tPA) was administered to patients following brain NCECT, in accordance with the established clinical guidelines.

### 2.3. Image Acquisition

All patients enrolled in the study underwent a NCECT at our public hospital using two different CT scanners (16 rows of detectors, 120 kV) of the same make and model (Phillips Ingenuity; Amsterdam, the Netherlands) during the diagnosis process of AIS. Patients were randomly assigned to each scanner. The acquisition and reconstruction protocols were identical in both scanners. The images obtained had a slice thickness of 0.625 mm. Although reconstructions with a thickness of 1 mm were available, they were not used for analysis. The window width and center were set at 80 and 40 Hounsfield units (HU), respectively (Figure 1).

### 2.4. Segmentation and Feature Extraction

The Digital Imaging and Communications in Medicine (DICOM) files of the brain NCECT were imported into 3D-Slicer version 5.2.2 (free software available at www.slicer.org) [23]. The region of interest (ROI) was defined as the visible thrombus in the ICA and/or proximal MCA (M1 and M2 segments). A 3D-semi-automatic segmentation was performed using the segment editor of 3D-Slicer [24]. The selected segmentation tool is “Level Tracing”, whereby the outline is defined by the movement of the mouse, resulting in pixels that share the same background density value as the mouse’s current background pixel. The subsequent application of the outline to the region of interest is a decision that is made by the radiologist. Segmentation was performed by a radiology resident who had received specialized training. The segmentation was validated by three neuroradiologists with expertise in neurointervention and neuroimaging (Figure 2). The RFs were also extracted in 3D-Slicer using the Radiomics module. For all ROIs, image voxel resampling to a dimension of 1 mm × 1 mm × 1 mm was performed. Smoothing with a Gaussian filter and a fixed gray bin width value of 25 was also performed to normalize the images. Wavelet-based features and kernel sizes of 3, 5 and 7 were also performed [25,26].

The RFs extracted were as follows: (1) first-order features (evaluate pixel/voxel distribution); texture features such as (2) gray-level cooccurrence matrix (GLCM), (3) gray-level dependence matrix (GLDM), (4) gray-level run length matrix (GLRLM), (5) gray-level size zone matrix (GLSZM) and (6) neighboring gray-tone difference matrix (NGTDM); (7) shape features (geometric evaluation of segmented area). A total of 106,197 radiomic features were extracted from the 91 patients included in the study, with 1167 features per patient (Figure 3).

The segmentation, extraction of RFs, and analysis of the results were performed using a system with an Intel CORE i7 processor (Santa Clara, CA, USA) with 16 GB RAM, 1 TB hard disk and Microsoft Windows 11 operating system (Redmond, Washington, DC, USA).

### 2.5. Clinical Data

This study also explored the relationship between various clinical variables and the achievement of FPE. The clinical variables analyzed included cardiovascular risk factors such as arterial hypertension, diabetes mellitus, dyslipidemia and smoking history, along with sex, age and thrombus laterality. In addition, the HU of each thrombus were collected and subjected to analysis.

### 2.6. Feature Selection and Statistical Analysis

The feature selection and the construction of the predictive model were carried out using Orange Data Mining Toolbox in Python (Ljubljana, Slovenia) and Statistical Package for the Social Sciences (SPSS) version 22.0 (SPSS Inc., Chicago, IL, USA) [27,28]. For the purpose of feature normalization, a Z-score normalization (with a range between 0 and 1) was applied using the tool “Continuize”, which is available in Orange. A LASSO (Least Absolute Shrinkage and Selection Operator) regression was used to select the significant RFs, with a nonzero coefficient. The RFs selected and the clinical variables were analyzed with a logistic regression, with Backward LR used as selection method. A confusion matrix, odds ratios and 95% confidence intervals were obtained. With regard to the confusion matrix, the true positive (TP) is defined as the number of FPE thrombectomies correctly identified as such. The false positive (FP) is defined as the number of non-FPE thrombectomies incorrectly identified as FPE thrombectomies. The true negative (TN) is defined as the number of non-FPE thrombectomies correctly identified as such. Finally, the false negative (FN) is defined as the number of FPE thrombectomies incorrectly identified as non-FPE thrombectomies (Table 1). The accuracy, sensitivity (Se) and specificity (Sp) of the model were calculated, where Se is the model’s ability to correctly classify patients with FPE thrombectomy and Sp is the model’s ability to correctly classify patients with non-FPE thrombectomy.

The performance of the model was evaluated using the receiver-operating characteristic (ROC) curve. The Hosmer–Lemeshow goodness-of-fit test was also calculated. Cohen’s Kappa coefficient (%) has also been calculated, which is defined for classification problems with two categories—in our case, FPE and non-FPE thrombectomies—as follows:K=100(Pa−Pe)/(1−Pe)
where $Pa=\left(TP+TN\right)/N$ and $Pe=\left(TP+FN\right)\left(TP+FP\right)/N^{2}+\left(FP+TN\right)\left(FN+TN\right)/N^{2}$. The Kappa index is a measure that adjusts for the effect of chance and is considered a more robust index than accuracy or AUC [29].

Finally, the Radiomic Quality Score (RQS) was calculated to measure the methodology compared with other radiomics studies. RQS consists of 36 checkpoints that assess radiomic studies to promote best scientific practice [30]. The score in our study was 17 (47.22%) (Appendix B). This present study was also elaborated by the CLEAR guidelines (CheckList for EvaluAtion of Radiomics research) (Appendix A) [31].

## 3. Results

### 3.1. Patient Information

A total of 279 patients with AIS underwent mechanical thrombectomy between January 2021 and May 2023. In 119 patients, the occlusion was in an artery other than the ICA/MCA, or a tandem occlusion. In 37 patients, the initial attempt at mechanical thrombectomy was conducted using the stent retriever or combined technique. In 32 cases, the precise location of the thrombus could not be determined using NCECT. Finally, 91 patients were included in the study (Figure 4).

Of the patients included, 51 achieved FPE while 40 did not. The cohort consisted of 31 male and 60 female patients. In the FPE group, 36 (70.6%) were female patients, 41 patients (80.4%) had arterial hypertension, 16 (31.4%) had diabetes, 34 (66.7%) had dyslipidemias, five (9.8%) had a history of smoking and 27 (53%) had clots located on the left side. In the non-FPE group, 24 (60%) were female patients, 30 patients (75%) had arterial hypertension, nine (22.5%) had diabetes, 22 (55%) had dyslipidemia, six (15%) had a history of smoking and 22 (55%) had clots located on the left side. The mean age was 77.74 years (standard deviation (SD) 12.13) in the FPE group and 77.70 years (SD 8.66) in the non-FPE group. The mean HU for the FPE group was 58.77 (SD 10.17) compared to 63.38 (SD 26.01) for the non-FPE group (Table 2).

### 3.2. Features Analysis

A total of 1167 RFs were obtained from each patient. The Least Absolute Shrinkage and Selection Operator (LASSO) method (λ = 0.9) selected nine RFs with nonzero coefficients. The RFs selected were one shape feature (maximum 2D diameter for rows), six texture features (GLRLM—gray-level non-uniformity, GLRLM—gray-level non-uniformity normalized, GLRLM—run entropy, GLSZM—zone entropy, GLSZM—size zone non-uniformity normalized, GLDM—large dependence low-gray-level emphasis) and two first-order features (90th percentile, Kurtosis). However, of the non-radiomics variables collected (gender, age, arterial hypertension, diabetes, dyslipidemia, smoking, laterality of clot and HU of clot) that were included in the logistic regression analysis, none were found to be statistically significantly associated with FPE. The association of these variables with FPE was also analyzed on an individual basis. The analysis of dichotomous variables was conducted using the Chi-square test. Continuous variables were analyzed using Student’s *t*-test. Nevertheless, the data did not demonstrate a statistically significant association between the acquisition of FPE through thromboaspiration and any of the aforementioned factors (*p*-value > 0.05) (Table 2). The Backward LR selection method of the logistic regression selected 6 RFs that were statistically associated with FPE (*p*-value < 0.05) for the final model prediction (Table 3).

### 3.3. Prediction of FPE

The nine RFs selected via LASSO regression (λ = 0.9), in conjunction with the clinical variables collected, were utilized to execute the logistic regression and thereby categorize the patients into FPE and non-FPE groups. The backward selection model of the logistic regression eliminates the clinical variables and three RFs (GLRM—run entropy, GLRLM—gray-level non-uniformity and GLDM—large-dependence low-gray-level emphasis) (*p*-value > 0.05). The final prediction model is exclusively predicated on the following radiomic features: 90th percentile, Kurtosis, maximum 2D diameter of rows, GLSZM—size zone non-uniformity normalized, GLSZM—zone Entropy and GLRLM—gray-level non-uniformity normalized (*p*-value < 0.005). With these six selected RFs, the confusion matrix and ROC curve were obtained, giving an accuracy of 0.813, a Se of 0.815, a Sp of 0.811 and an AUC of 0.890 (*p* < 0.05) (Table 4) (Figure 5).

The Kappa coefficient result was 74.90%, indicating a substantial agreement between the prediction model and the actual observations (Table 5). For the Hosmer–Lemeshow goodness-of-fit test, the Chi-square test result was 12.002 with a *p*-value of 0.151. This means that there is no reason to believe that the predicted results are different from the observed results and the model can be considered acceptable (Table 6).

## 4. Discussion

The objective of this study was to predict the FPE in patients with MCA/ICA stroke who were undergoing MT by thromboaspiration. This was achieved by utilizing RFs obtained from the semi-automated segmentation of the hyperdense thrombus on the NCECT. The results demonstrate that radiomic data can help to predict FPE in these patients. Six RFs were independent predictors for FPE prediction (*p*-value < 0.05). The accuracy of FPE prediction is 0.813 when these six RFs are employed. The clinical data obtained and the clot density were not demonstrated to be independent predictors for FPE (*p*-value > 0.05). The findings of this study suggest that thrombi visible on NCECT contain information associated with the efficiency of thrombus removal by thromboaspiration.

Forecasting the likelihood of achieving FPE through thromboaspiration could be crucial in selecting the most cost-effective method for arterial reperfusion. The effectiveness of different thrombectomy techniques remains a topic of debate. Both thromboaspiration and a stent retriever show similar outcomes in large vessel occlusions [32]. Many hospitals use thromboaspiration as the primary technique, reserving stent retrievers for second-line use when thromboaspiration fails or as a first-line option for distal occlusions in smaller vessels. Few studies have directly compared the cost-effectiveness of these techniques. Although a procedure that begins with thromboaspiration and subsequently requires stent retrieval is generally less cost-effective than starting with a stent retriever, higher cost-effectiveness has been observed when FPE is achieved with a single pass of thromboaspiration compared to using a stent retriever [33].

The relationship between radiomic features and successful permeabilization in AIS patients has been previously investigated. Hofmeister et al. developed a radiomic model to predict TICI > 2b with a single pass of thromboaspiration in a cohort of 47 patients [20]. The model demonstrated an accuracy of 0.851, a sensitivity of 0.50, a specificity of 0.971, a positive predictive value of 0.857 and a negative predictive value of 0.850. A Support Vector Machine (SVM) classifier was employed, and the model was based on 1485 radiomic features (first-order features, GLCM, GLDM, GLRLM, GLSZM, NGTDM and shape features), with nine selected for the final prediction model. Additionally, this study identified a correlation between radiomic features and the number of thrombectomy attempts. Despite the smaller size of their cohort, their findings suggest a correlation between radiomic data and the success of mechanical thrombectomy. In contrast to our study, which aims to predict FPE (TICI ≥ 2c), they defined successful reperfusion as TICI ≥ 2b. Hofmeister et al. [20] employed semi-automated segmentation and a SVM automatic classifier, whereas our study achieved optimal results with the logistic regression analysis. The previously mentioned study reported a radiomic quality score of 17 out of 36 (47.22%), which is equivalent to this present study’s score. Sarioglu et al. [21] conducted another similar study. They developed a prediction model for FPE in 52 patients treated primarily with stent retrievers. Their model achieved an AUC of 0.83 in predicting FPE and demonstrated that clot-based radiomics can aid in estimating the success of mechanical thrombectomy in AIS patients. They performed a manual segmentation of thrombi on non-contrast CT and validated the thrombus location using CT angiography, extracting 88 radiomic features. They identified two RFs as independent predictors of FPE and found that incorporating these features into a model based on ASPECTS and patient sex improved prediction accuracy. The findings of their study also emphasize the significance of RFs in the planning of endovascular procedures. Nevertheless, they attempted to pre-predict FPE with a stent retriever, their sample size was smaller, and the number of features they employed was also smaller than that of the present article. Furthermore, they utilized manual segmentation, which resulted in limitations in terms of the reproducibility of ROI segmentation [21].

Our study has several limitations. First, only patients with visible thrombi on NCECT were included in this study. Although the relationship between thrombus density and the likelihood of achieving FPE has been investigated, no statistically significant association was found [34]. Similarly, an analysis of thrombus density in this present article revealed no significant correlation between mean HU and FPE. For practical integration into clinical settings, it is essential that the impact of these tools on image analysis time be minimized. The segmentation of visible thrombi in NCECT is quick and straightforward, whereas including non-visible thrombi on NCECT can significantly delay analysis when localization is based on angio-CT. Future research should explore the inclusion of non-visible thrombi at NCECT by integrating additional imaging techniques for segmentation. Secondly, this is a retrospective study conducted at a single center, so it does not present an external validation cohort. Despite the fact that this study encompasses patients whose images were obtained using two different CT scanners—both scanners are of the same make and model and utilized the same acquisition and reconstruction protocol—this does not enhance the external validity of the article. In this context, we suggest that multicenter and prospective studies be conducted to broaden the current body of literature. Third, the lack of standardization in the procedures for obtaining and analyzing radiomic data remains a limitation in applying findings to other research groups. Therefore, it is crucial to provide a detailed description of the methodology used and adhere to established standards for radiomic studies, such as the Radiomics Quality Score (RQS) and CLEAR checklist. Finally, with regard to the issue of sample size, while this manuscript includes a greater number of patients than similar published studies [18,19], and while it is a retrospective study, the ideal sample size calculated for a prospective study should be 162 patients (taking into account a population of 279 mechanical thrombectomies performed in the recruitment period, with a confidence interval of 95% and a margin of error of 5%). The inclusion of additional patients was precluded by the relatively brief recruitment period of just sixteen months. Furthermore, the inclusion and exclusion criteria were meticulously designed to exclude patients with some conditions that could impede the successful acquisition of a FPE during mechanical thrombectomy (e.g., patients with tandem occlusion or distal occlusions). In this sense, despite the study’s larger sample size compared to other similar published studies, further increases in the number of patients are necessary to ensure the robustness of the findings.

Currently, the decision to perform mechanical thrombectomy using either aspiration or a stent retriever is guided by the neuroradiologist’s personal preference, the patient’s vascular anatomy and the location of the thrombus but does not usually consider the thrombus composition. Radiomics provides additional insights into the radiological appearance of the thrombus, which is closely related to its composition [35,36]. This study has shown that radiomic data can provide valuable information for predicting FPE with thromboaspiration. By integrating this information, neuroradiologists could better predict the likelihood of success of mechanical thrombectomy and select the most appropriate technique for each case, if targeted studies support these findings. This approach would transform mechanical thrombectomy into a more personalized technique.

## 5. Conclusions

Our study demonstrates that radiomic features can help predict the achievement of FPE with thromboaspiration, potentially improving procedural planning and allowing for a more tailored approach to mechanical thrombectomy.

## Figures and Tables

**Figure 1 jcm-14-04027-f001:**
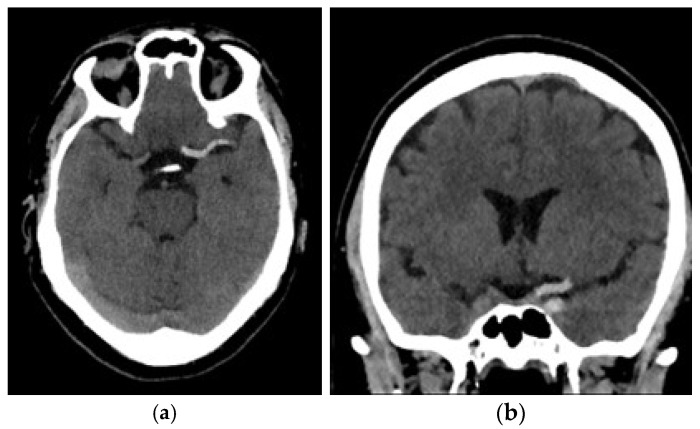
Example of AIS caused by a thromboembolus in the M1 segment of the left middle cerebral artery: (**a**) Sagittal view; (**b**) coronal view.

**Figure 2 jcm-14-04027-f002:**
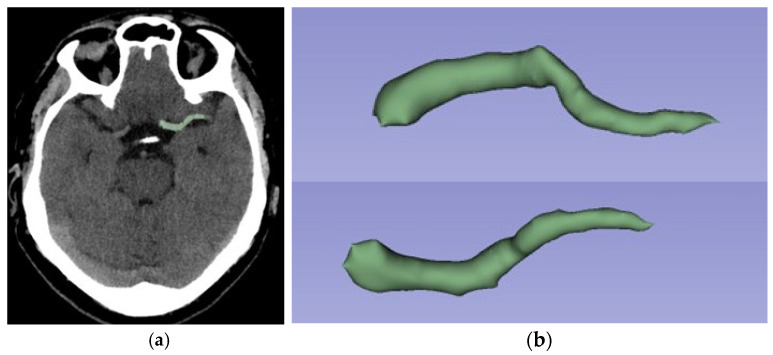
Same patient as in Figure 1. Example of semi-automated segmentation using 3D Slicer software: (**a**) Sagittal view segmented; (**b**) 3D reconstruction of thrombus segmentation. The selected segmentation tool is “Level Tracing”.

**Figure 3 jcm-14-04027-f003:**
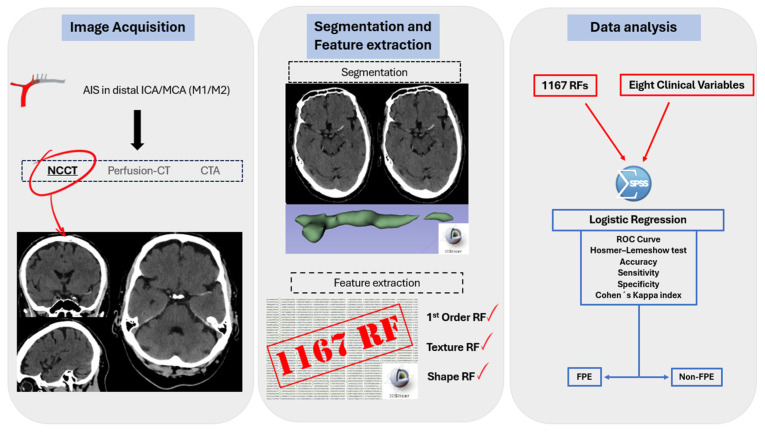
Radiomics workflow: RFs are obtained from visible thrombus in NCECT. Segmentation is semi-automatic. A total of 1167 radiomic features were obtained for each patient, with the most significant ones selected using LASSO regression. Finally, a logistic regression identified six statistically significant RFs for the prediction model. The accuracy, sensitivity, specificity, ROC curve and Cohen’s Kappa index were then calculated.

**Figure 4 jcm-14-04027-f004:**
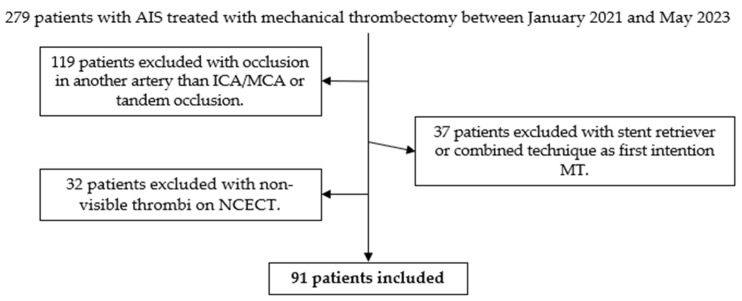
Decision tree for inclusion of patients in the study. From a total of 279 patients, 91 patients were selected after applying the inclusion and exclusion criteria.

**Figure 5 jcm-14-04027-f005:**
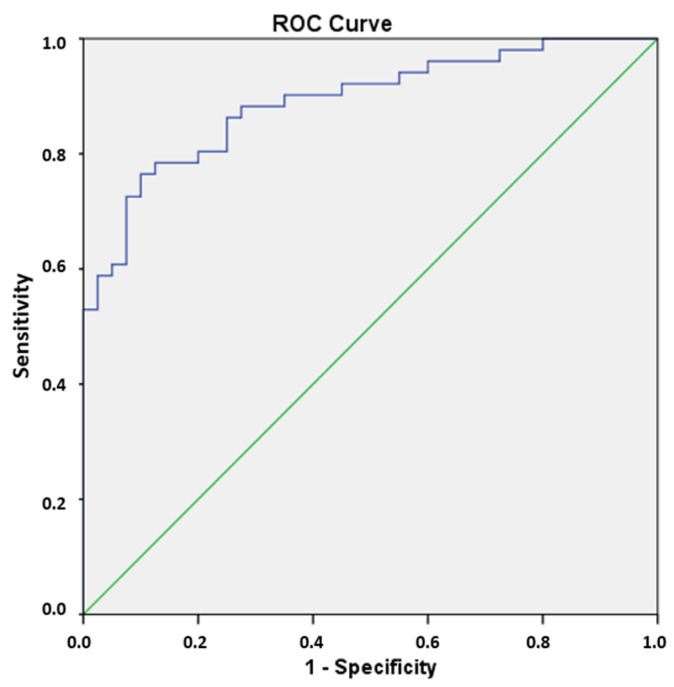
The ROC curve was obtained with SPSS using the six RFs selected in the logistic regression. The green line represents an AUC of 0.5. The blue line shows the AUC obtained in the FPE prediction (0.890).

**Table 1 jcm-14-04027-t001:** Representation of a confusion matrix used to visualize the performance of the prediction of FPE. Columns represent the predicted class. The rows represent the true class.

		Predicted	
		FPE	Non-FPE
**Observed**	**FPE**	TP *	FP *
	**Non-FPE**	FN *	TN *

* TP: true positive. TN: true negative. FP: false positive. FN: false negative.

**Table 2 jcm-14-04027-t002:** Baseline characteristics of patients included in this study.

Patient Data	FPE	Non-FPE	*p*-Value
Sex	Female: 36 (70.6% ^1^)	Female: 24 (60%)	0.290
Age (mean)	77.74 (SD ^2^ 12.13)	77.70 (SD 8.66)	0.559
Hounsfield Units (mean)	58.77 (SD 10.17)	63.38 (SD 26.01)	0.296
Arterial Hypertension	41 (80.4%)	30 (75%)	0.538
Diabetes	16 (31.4%)	9 (22.5%%)	0.347
Dyslipidemia	34 (66.7%)	22 (55%)	0.256
Smoke	5 (9.8%)	6 (15%)	0.370
Laterality	Left: 27 (53%)	Left: 22 (55%)	0.845

^1^ Percentage represents proportion of patients with condition in group (FPE or non-FPE). ^2^ SD: standard deviation.

**Table 3 jcm-14-04027-t003:** The LASSO regression identified nine RFs with non-zero coefficients. Logistic regression showed that six of these nine RFs were statistically associated with the likelihood of achieving FPE with thromboaspiration.

RFs	Class of RF	OR ^1^	*p*-Value
90th Percentile	First order	0.809	0.045
Kurtosis	First order	0.536	0.020
Maximum 2D Diameter Row	Shape	0.646	0.004
Size Zone Non-Uniformity Normalized	GLSZM	1.848 × 10^−4^	0.010
Zone Entropy	GLSZM	0.205	0.006
Gray-Level Non-Uniformity Normalized	GLRLM	0.06	0.005
Run Entropy	GLRLM	0.268	0.605
Gray-Level Non-Uniformity	GLRLM	1.383	0.240
Large Dependence Low-Gray-Level Emphasis	GLDM	1.003	0.317

^1^ OR: odds ratio.

**Table 4 jcm-14-04027-t004:** Confusion matrix obtained from logistic regression analysis using the six RFs associated with the likelihood of achieving FPE with thromboaspiration.

		Predicted		∑
		FPE	Non-FPE	
**Observed**	**FPE**	44	7	51
	**Non-FPE**	10	30	40
**∑**		54	37	**91**

**Table 5 jcm-14-04027-t005:** Interpretation of the Kappa index. The results of our model show substantial agreement.

Kappa Value (%)	Interpretation
0	Agreement equal to chance
10–20%	Slight agreement
21–40%	Fair agreement
41–60%	Moderate agreement
61–80%	Substantial agreement
81–99%	Near-perfect agreement
100%	Perfect agreement

**Table 6 jcm-14-04027-t006:** Hosmer–Lemeshow contingency table.

Step	Non-FPE		FPE	
	Observed	Predicted	Observed	Predicted
1	8	8.899	1	0.101
2	8	8.073	1	0.927
3	6	6.576	3	2.424
4	7	5.066	2	3.934
5	5	3.939	4	5.061
6	3	2.859	6	6.141
7	2	2.007	7	6.993
8	1	1.259	8	7.741
9	0	0.823	9	8.177
10	0	0.499	10	9.501

## Data Availability

The software utilized for segmentation and feature extraction can be made accessible via https://www.slicer.org/. Patient images and radiomic data are not published for ethical reasons.

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
