# Peer review of "Is Personalized Mechanical Thrombectomy Based on Clot Characteristics Feasible? A Radiomics Model Using NCECT to Predict FPE in AIS Patients Undergoing Thromboaspiration"

_jcm, 2025, doi:10.3390/jcm14124027_

Round 1

Reviewer 1 Report

Comments and Suggestions for Authors

Dear Authors,
I reviewed your manuscript.
I carefully evaluated the study and appreciated the relevance of the topic and the methodological effort. The development of a radiomics-based model to predict First Pass Effect (FPE) in acute ischemic stroke patients undergoing thromboaspiration is clinically meaningful and timely.

My question is: Why haven't you developed a machine learning model? I see that in the discussion you cite, works chw have followed this approach. What was the rationale for not developing a training and testing of your own prediction model?
The following are some suggestions aimed at improving the clarity, scientific accuracy and overall quality of the manuscript:

In section 2.3, correct the typo “adquisition” to “acquisition.”

Clarify that although two different scanners were used, the acquisition and reconstruction protocols were identical to reinforce the homogeneity of imaging data.

Consider adding a reference to PMID: 33826964 in the introduction to support the concept of applications of deep learning in stroke imaging and outcome prediction.
Figure 4 looks cut off and is currently not readable; please revise the layout of the figure to ensure that all content is visible.

In Table 3, replace the term “RF Type” with “RF Class". Make OR explicit in the table legend.

The manuscript does not address the lack of external validation; this should be explicitly addressed both in the discussion and in the sections on limitations.

Overall, the manuscript is well structured and presents a sound methodology. Implementation of these suggestions will improve the scientific rigor and reproducibility of the study.

Thanks & Regards.

Author Response

Dear reviewer,
The authors of this article would like to thank you for your time and valuable comments.
In response to your queries:

Question 1: Why haven't you developed a machine learning model? I see that in the discussion you cite, works chw have followed this approach. What was the rationale for not developing a training and testing of your own prediction model?

Initially, the sample was divided into a training group and a test group, with 70% of the sample allocated to the training group. The issue we encountered was that the size of the test group was inadequate (27 patients). We evaluated different automatic classifiers (SVM, Naive Bayes, Random Forest, Neural Network, and Logistic Regression) using the Orange: Data Mining toolbox in Python. The results were found to be highly comparable, with an Area Under the Curve (AUC) and Accuracy of approximately 0.800.

Due to the limited size of the test group, we ultimately opted to perform cross-validation with LOOCV to avoid any division of the group (a system with which we are already acquainted from previous publications). We are currently recruiting a cohort for our research in order to develop a model with training and test groups that have an adequate sample size. 

We would like to thank you for your interesting question.

With regard to the remaining comments:

In section 2.3, correct the typo “adquisition” to “acquisition.” Corrected, thank you very much.

Clarify that although two different scanners were used, the acquisition and reconstruction protocols were identical to reinforce the homogeneity of imaging data. Added, thank you very much (lines 106 and 320).

Consider adding a reference to PMID: 33826964 in the introduction to support the concept of applications of deep learning in stroke imaging and outcome prediction. Added (reference 16, line 61). Very interesting article, thank you very much.

Figure 4 looks cut off and is currently not readable; please revise the layout of the figure to ensure that all content is visible. Checked, thank you very much.

In Table 3, replace the term “RF Type” with “RF Class". Make OR explicit in the table legend. Checked, thank you very much.

The manuscript does not address the lack of external validation; this should be explicitly addressed both in the discussion and in the sections on limitations. Added (fourth paragraph of the discussion, lines 317-322), thank you very much.

Finally, we would like to thank you again for your time and valuable help in improving our manuscript. We hope that the changes implemented will be to your satisfaction.
Best regards from all the authors.

Reviewer 2 Report

Comments and Suggestions for Authors

This paper mainly studies how to use the radiomics model based on non-contrast-enhanced computed tomography (NCECT) to predict whether patients with acute ischemic stroke (AIS) can achieve the First Pass Effect (FPE) when undergoing thrombus aspiration.my comments are as follows:

  1. A total of 91 patients were included in the article, but it was not mentioned whether the sample size was sufficient to support the conclusion of the statistical analysis. In the discussion section, the influence of sample size on the research results should be discussed in detail
  2. The article does not elaborate on the specific criteria for patient selection, which may lead to the heterogeneity of the sample.  It is recommended to clarify the inclusion and exclusion criteria for patients to ensure the reliability of the research results.doi:10.1007/s00330-025-11419-1;10.3389/fimmu.2024.1446511;The approaches in the above two articles are worthy of the author's learning and should be cited in the article. And it is suggested that Figure 2 and table1 be redrawn according to the given literature 2(important)->doi:10.3389/fimmu.2024.1446511;
  3. In the image preprocessing section, it was mentioned to use Gaussian filtering and a fixed gray level width value for normalization, but the basis for the selection of these parameters was not explained.
  4. In LASSO regression analysis, the selection of regularization parameters is crucial to the feature screening results. The determination method of regularization parameters is not mentioned in the text. It is suggested to supplement the relevant description and discuss its influence on feature selection.
  5. Although it was mentioned in the text that there was no statistically significant association between clinical variables and FPE, the interactions among these variables were not analyzed in detail.
  6. The confusion matrix is an important tool for evaluating the performance of the model. However, in this paper, only the content of the matrix is listed, and the various indicators in the matrix (such as true examples, false positive examples, etc.) are not explained in detail.
  7. The article mentions a comparison with the research of Hofmeister et al., but does not delve into the differences between the two in terms of research design, feature selection and model performance.

Author Response

Dear reviewer,

We would like to thank you for taking the time to review and improve our manuscript, as well as for the valuable comments you have sent us.
With regard to the points that have been identified for improvement of our manuscript:

Comment 1 [A total of 91 patients were included in the article, but it was not mentioned whether the sample size was sufficient to support the conclusion of the statistical analysis. In the discussion section, the influence of sample size on the research results should be discussed in detail]

Please refer to the fourth paragraph of the discussion, where we have added an additional limitation. We conducted a sample calculation as if it were a prospective study, which indicated a sample size of 162 patients. It is not feasible to reach the stipulated number of patients, as we must exclude those for whom the location of the thrombus may hinder thrombectomy (tandem occlusions and distal occlusions). This approach is consistent with other studies that investigate factors associated with mechanical thrombectomy. It is also worth noting that published studies similar to ours [20 and 21] have similar or smaller sample sizes than our manuscript. We hope you will find it to your satisfaction, but if not, we are at your disposal to modify any aspect you consider necessary. We would like to express our sincere gratitude for your appreciation.

Comment 2: [The article does not elaborate on the specific criteria for patient selection, which may lead to the heterogeneity of the sample.  It is recommended to clarify the inclusion and exclusion criteria for patients to ensure the reliability of the research results.doi:10.1007/s00330-025-11419-1;10.3389/fimmu.2024.1446511;The approaches in the above two articles are worthy of the author's learning and should be cited in the article. And it is suggested that Figure 2 and table1 be redrawn according to the given literature 2(important)->doi:10.3389/fimmu.2024.1446511;]

We would like to express our gratitude for your contribution to this matter. Please note that the inclusion criterion '(1) Patient aged 18 years old or older' has been added in the second paragraph of section 2.2 Patient Selection, as they do in the bibliography provided by the reviewer. Please refer to the rest of the inclusion and exclusion criteria below. Figure 4 provides a detailed breakdown of the number of mechanical thrombectomies performed during the recruitment period, along with the number of patients who did not meet the inclusion and exclusion criteria. The word 'excluded' has been added to clearly identify patients excluded because they did not meet the inclusion/exclusion criteria. The inclusion/exclusion criteria were established in accordance with other studies also examining the performance of mechanical thrombectomy. Furthermore, we sought to conduct studies with high image quality, which is why one of the inclusion criteria is that the CT images have a thin slice thickness.

We would like to express our gratitude for the recommended articles, which we found to be very interesting and have therefore added to the bibliography (references 15 and 16).

With regard to the redrawing of table 1 and figure 2: table 1 details the confusion matrix, and its format cannot be altered because it must follow the format of the journal. Please refer to Figure 2, which illustrates the semi-automatic segmentation process. Please note that the layout and format of the article have been adapted to align with the journal's specific requirements.

Comment 3: [In the image preprocessing section, it was mentioned to use Gaussian filtering and a fixed gray level width value for normalization, but the basis for the selection of these parameters was not explained.]

We would like to express our gratitude for your inquiry, which we found to be very interesting. This has been one of the main challenges during the preparation of the article, as none of the similar published studies provided their pre-processing data. For this reason, the recommendations of S. Mehta et al. and Shipitko et al. have been followed. The references of these articles have been added to the bibliography (references 25 and 26).

Comment 4: [In LASSO regression analysis, the selection of regularization parameters is crucial to the feature screening results. The determination method of regularization parameters is not mentioned in the text. It is suggested to supplement the relevant description and discuss its influence on feature selection.]

We appreciate your feedback and have included the lambda used for feature selection in the LASSO regression (second line of section "3.2 Features analysis" and first line of section "3.3 Prediction of FPE").

Comment 5: [Although it was mentioned in the text that there was no statistically significant association between clinical variables and FPE, the interactions among these variables were not analyzed in detail.]

Thank you for your input. We add a paragraph with an analysis of the clinical variables included and the derivation of the PEF (in "3.2 Features analysis", at the end of the paragraph). Dichotomous clinical variables were analysed with the X2 test. Continuous variables were analysed with the Student's t-test. None of them showed a statistically significant relationship. We update table 2 of baseline characteristics of the patients included in the study with the p-values obtained in these analyses. Thank you very much.

Commentary 6: [The confusion matrix is an important tool for evaluating the performance of the model. However, in this paper, only the content of the matrix is listed, and the various indicators in the matrix (such as true examples, false positive examples, etc.) are not explained in detail.]

Thank you for your commentary. Please refer to the end of Section 2.7, 'Feature selection and statistical analysis', where you will find a detailed interpretation of the confusion matrix. The meaning of TP, TN, FP and FN is explained in full in our work. Please refer to Table 1, which is a representation of the confusion matrix, formatted in accordance with the journal's requirements. Please find below a comment that has been added to the bottom of table 1.

Commentary 7: [The article mentions a comparison with the research of Hofmeister et al., but does not delve into the differences between the two in terms of research design, feature selection and model performance.]

Thank you for your feedback. The comparison with the Hofmeister et al. study is made in section 4 "Discussion". These comparation begins on line 280 and concludes on line 295. The commentary provides a comprehensive overview of the number of patients, the segmentation method, the automatic classifier used, the number of features extracted, the results and the RQS. We have incorporated the radiomic variable classes used in the study. Finally, we compare these aspects with our research, detailing the main differences between the two studies.

Finally, we would like to thank you for taking the time to improve our manuscript. We hope that you will be pleased with the changes that have been implemented. If not, we will modify the article as you see fit.
We are pleased to extend a warm greeting from the authors.

Round 2

Reviewer 2 Report

Comments and Suggestions for Authors

thank you for your revisions.i have no further comments